

# IoT-CCAC: a blockchain-based consortium capability access control approach for IoT

Mohammed Amine Bouras[1], Boming Xia[1], Adnan Omer Abuassba[2], Huansheng Ning[1] and Qinghua Lu[3]

[1] School of Computer and Communication Engineering, University of Science and Technology Beijing, Beijing, China
[2] IT Department, Arab Open University, Al-Bireh, Palestine
[3] Data61, the Commonwealth Scientific and Industrial Research Organisation CSIRO, Sydney, NSW, Australia

## ABSTRACT

Access control is a critical aspect for improving the privacy and security of IoT systems. A consortium is a public or private association or a group of two or more institutes, businesses, and companies that collaborate to achieve common goals or form a resource pool to enable the sharing economy aspect. However, most access control methods are based on centralized solutions, which may lead to problems like data leakage and single-point failure. Blockchain technology has its intrinsic feature of distribution, which can be used to tackle the centralized problem of traditional access control schemes. Nevertheless, blockchain itself comes with certain limitations like the lack of scalability and poor performance. To bridge the gap of these problems, here we present a decentralized capability-based access control architecture designed for IoT consortium networks named IoT-CCAC. A blockchain-based database is utilized in our solution for better performance since it exhibits favorable features of both blockchain and conventional databases. The performance of IoT-CCAC is evaluated to demonstrate the superiority of our proposed architecture. IoT-CCAC is a secure, salable, effective solution that meets the enterprise and business's needs and adaptable for different IoT interoperability scenarios.

## INTRODUCTION

As we step into the Internet of Things (IoT) era where ubiquitous objects are connected, the number of IoT devices has witnessed an unprecedented increase. According to Juniper Research, there will be more than 46 billion IoT devices in 2021 (*Juniper Research, 2016*). The proliferation of the IoT has brought many benefits to us, boosting various technologies such as smart home (*Dhelim et al., 2018*) and smart city (*Camero & Alba, 2019*). However, both current and future IoT systems also cause concerns in terms of security and privacy (*Xu et al., 2018b*). Specifically, malicious users may gain access to devices that do not belong to them, deliberately tamper data, and even steal valuable information. As a countermeasure, access control for IoT has been a popular research topic

Corresponding author
Mohammed Amine Bouras,
bouras.ma@xs.ustb.edu.cn

and a crucial aspect of IoT security and privacy (*Singh et al., 2015*; *Ouaddah et al., 2017*; *Bouras et al., 2020*).

Conventional access control methods (e.g., role-based access control (RBAC), attribute-based access control (ABAC), capability-based access control (CBAC)) have been widely applied to IT systems (*Xu et al., 2018b*). Compared to the two schemes, CBAC is relatively more lightweight as it uses a communicable and unforgeable token of authority, which associates an object with corresponding access rights. However, one drawback of the original CBAC is that a token can only be granted to one subject, which may cause low efficiency and calls for a proper solution. Also, these access control methods mostly rely on centralized solutions, which may lead to several problems. Firstly, central management may end up with single-point failures because many systems suffer from security issues related to the tools used to manage the platforms. Secondly, the reliance on a central server or a third party gives them access to perform checks on stored data, which could lead to privacy leakage. Third, such centralized system are not designed for a consortium applications as the transparency is omitted.

Blockchain keeps all transaction records through a peer-to-peer network as a distributed ledger. It is essentially a growing list of records (i.e., blocks) linked to the previous block via cryptography. Blockchain possesses various features (e.g., decentralization, tamper-proof, security) that make it a trustable alternative infrastructure for access control systems. Thus, when integrated with the blockchain technique, access control can bring the following favorable advantages: (a) help eliminate third parties, solve single-point failures and other centralized management problems; (b) have access to trustable and unmodifiable history logs; (c) consensus mechanisms are applied that only valid transactions are recorded on the blockchain; (d) smart contracts can help monitor and enforce access permissions under complex conditions.

However, the use of blockchain for IoT access control also comes with some limitations. First of all, blockchain is not designed to store a significant volume of data, which usually requires the proper integration of on-chain and off-chain databases to handle specific tasks. Second, the transactions in public blockchain can be viewed by anyone which does not accommodate the need of a consortium enterprise network because its transactions must be private and only accessible to consortium members. Although private blockchain (e.g., Ethereum private blockchain, Hyperledger Fabric) has been developed to solve this problem, it is not the only viable solution—a blockchain database (e.g., BigchainDB (*McConaghy et al., 2016*)) can do the same with the even better performance (*Tseng et al., 2020*). Third, performance and scalability have always been two significant problems of blockchain technology. Regardless transaction execution and validation performance has been improved recently by introducing lighter consensus mechanisms (*Biswas et al., 2019*), and more efficient transaction scheme such as Hyperledger Fabric (*Androulaki et al., 2018*), the performance and scalability of the blockchain-based access control solutions still cannot compete with the current centralized solutions.

Therefore, based on the aforementioned limitations of existing blockchain-based access control methods, here we present an enhanced blockchain-based capability access control architecture for IoT named IoT-CCAC, IoT Consortium Capability-based Access

Control Model. In our design, we focus on interoperability and data exchange by organizing the access control data in form of assets (physical devices), services (collaborative applications), and profiles (the representation of the asset inside a service) to make the solution granular and flexible taking in consideration fast growing and the scalability of IoT. In addition, we introduce the concept of statement, which can be granted to a subject or a group of subjects as a single capability token or group capability token. Different from the other IoT capability based access control methods, our solution is designed for consortium networks instead of personal networks. Based on the aforementioned limitations of blockchain, we further investigate the blockchain based database that combines the security properties of blockchain and the performance advantage of a database and use it as a backbone of the proposed access control. The contributions of this paper mainly include:

- Faced with the centralized problem of most existing IoT access control methods and the limitation of current blockchain-based solutions, we present an enhanced decentralized capability-based access control architecture for consortium applications named IoT-CCAC.
- The notation of the group capability token is introduced as a measure to improve the conventional capability-based solutions and works.
- We discuss the IoT access control data registry requirements, and we present the blockchain-based database integration architecture.
- The proposed approach is implemented and evaluated in proof-of-concept prototype. The results shows IoT-CCAC is fast, secure and can scale and support IoT city and business applications.

The remainder of this paper is organized as follows. "Related Works" presents related works of blockchain-based IoT capability access control solutions. "IOT Consortium Capability-based Access Control Model (IOT-CCAC)" presents the IoT-CCAC architecture and define it's components, token generation protocol and authorization scheme. "IOT-CCAC and Blockchain Integration" discusses the requirements of IoT access control data registry and the blockchain-based database integration. In "Implementation and Evaluation", we implement and evaluate the prototype of our proposed approach and discuss it's security and performance aspects. We complete our work with a conclusion and an outlook for the future and following works.

## RELATED WORKS

In this section, we mainly summarize some research on the integration of blockchain and CBAC for IoT. Specifically, CBAC is selected considering its relative advantages over RBAC and ABAC. For instance, by leveraging CBCA, a subject can complete its task using the minimum of access rights (i.e., the principle of least privilege) (*Nakamura et al., 2019*). The detailed comparison of the three access control methods is summarized below (see Table 1) in terms of their corresponding explanation, scalability, heterogeneity, dynamicity, lightweight, flexibility, and granularity.

**Table 1 Comparison of three access control methods (*Ouaddah et al., 2017*).**

| AC approach | Role-based AC | Attribute-based AC | Capability-based AC |
|---|---|---|---|
| Description | Employs pre-defined roles that carry a specific set of privileges. To grant access you have to give the object a role | Uses policies which are defined according to a set of selected attributes from the user, subject, resource, and environment attributes and so on | Uses a communicable, unforgeable token of authority. The token references an object along with an associated set of access rights |
| Scalability | Not scalable as pre-defining roles for billions of devices is not possible and will drive to many errors when assigning roles to fast-changing devices | The access policies are defined on attribute which gives it the scalability feature because in a complex system or nested policies the more granular your system is the more is efficient to handle billions of devices | Scalability is made possible by providing tokens only (the management of tokens are easier and efficient), but it can be a problem for complex systems (many components) where a user may handle tens of tokens where each token represents an access right |
| Heterogeneity | Moderate | High | High |
| Dynamicity | Low | High | Moderate |
| | (A role is not dynamic as it's pre-defined and changing a role will affect all the associated devices) | (The access policies are defined by a set of conditions which makes it dynamic and more robust to changes) | (every time I change the policy I need to change the token) |
| Lightweight | Moderate | Moderate | High |
| Flexibility | Moderate | High | High |
| Granularity | Low | High | Moderate |

## Blockchain-based capability access control for IoT

Abundant work has been carried out on the topic of integrating IoT access control with blockchain. There exists much research on applying CBAC to IoT (*Ouaddah et al., 2017*) considering its characteristics such as lightweight and scalability, and these features also make it a preferred choice to be integrated with blockchain to provide more secure access management for IoT. However, only a few existing studies have explored the potential of combining CBAC with blockchain-related technology to manage IoT identity management and access control and all works were designed for IoT personal networks.

*Xu et al. (2018a)* propose a complete blockchain-enabled CBAC strategy for IoT called BlendCAC. Then, in another work (*Xu et al., 2019*), the authors further modify BlendCAC in the case of space situation awareness to handle identity authentication via a virtual trust zone, token management, and access right validation. To evaluate the feasibility of BlendCAC, experiments are carried out on a private Ethereum blockchain and demonstrated its effectiveness. However, the capabilities of subjects and their delegation relationships are managed by using a delegation tree in BlendCAC, which can cause incomplete recorded delegation information. Also, two types of tokens in BlendCAC must be consistently updated, which cannot always be met. In addition, the BlendCAC is partially decentralized as it employs a cloud server to coordinate between the domains and to be the service provider.

To address the delegation problem in BlendCAC, *Nakamura et al. (2019)* introduced the delegation graph in place of the delegation tree. Moreover, Ethereum smart contracts were used for the storage and management of capability tokens. Later, they further

enhance the method and propose to handle token management according to its actions or access rights instead of conventionally used subjects (*Nakamura et al., 2020*). However, the work is still lack of systematic architecture design meeting the IoT requirements. For example, the work focuses on solving the problems of delegation ambiguity without taking in consideration that in a personal network issuing large number of tokens without a solid management will cause the ambiguity to system users.

However, the above CBAC studies suffer also from the lack of organization and management of information inside the system. For instance, a network of massive connected sensors and devices will raise the problem of data management and classification which will lead to traceability and analysis issues and slow the process of continues security enhancement. In addition, the proposed works don't support interoperability and data exchange between the IoT domains and organization as the solution is proposed for a personal IoT network and it doesn't fit the city or business IoT network and applications.

Comparing to existing work, this study aims to provide a fine-grained, scalable and high performance CBAC solution for IoT city and business consortium networks. We designed a modular CABC system to enhance flexibility of the solution by defining and creating a framework for the transactions and data. The design decision adopted enables interoperability and data exchange between the network members and impose the principal of least privileges.

# IOT CONSORTIUM CAPABILITY-BASED ACCESS CONTROL MODEL (IOT-CCAC)

In this section, we design and overview the essential aspects adopted in this work for an IoT consortium capability-based access control model. We also give a detailed description of the linkages between all the components presented in our proposal.

## IoT-CCAC description

IoT access control is a paradigm of defining policies and assigning them to users, groups of users, and network resources such as devices and sensors defining their permissions and protecting the network from malicious and unauthorized access. For instance, IoT is a complex network of connected domains where each domain has its sub-network, and each sub-network manages its resources. Defining policies for a complex network depends on the degree of flexibility, granularity, and privacy maintained in the ecosystem, considering the interoperability and cross-organizational information exchange. Therefore, IoT-CCAC allows every domain to define, manage, and share its resources to enable interoperability in services with other organizations and hand the control of the network and sub-networks resources to its owners. To better illustrate the proposed model, we define relevant IoT network and IoT-CCAC components as presented in Table 2.

## Identity management external component

Identity management (IDM) is a crucial feature of any digital environment, especially IoT ecosystem access control. Each IoT entity must have a unique identifier representing its

**Table 2 IoT-CCAC terms and descriptions.**

| Term | Description |
|------|-------------|
| Domain | a member of group of organization participating in the consortium network |
| Subject | a human user or a device that interacts with the consortium network and applications |
| Resource | an entity as a service in the network, such as a temperature sensor or a document data |
| Asset | the digital representation of a physical resource owned by a participating domain |
| Service | a service or an application initiated by several domains under a collaborative project |
| Profile | the representation of an asset inside a service |
| Context | environmental information gathered from resources, such as location and time |
| Statement | a document defines the access rights granted to a subject to access a resource |

identity. The IDM typically has three main functions, which are registration, authentication, and revocation. Registration to upload an entity identity to the system and assign a unique identifier, authentication to inspect an entity identity each time reacts with the ecosystem, and revocation to withdraw the digital identity of an entity (*Bouras et al., 2020*). In our design. All the aspects related to the authentication are out of scope for this work.

## IoT-CCAC system architecture

The main components of the IoT-CCAC system are asset management, service management, profile management, context management, and statement management. The system also has a token verification module and a unique identifier (UID) generator module, as shown in Fig. 1.

### *Asset management*

The asset management allows each domain to register and store its physical resources in the form of assets, and only the asset owner can edit or withdraw its own asset. The system assets are the available physical resources that services can use and interact with. Assets are used mainly for network resource discovery, classification, and other modeling strategies and digital representation. Properties needed for creating an asset can be expressed with the following notations:

$$Asset = \{assetConext, assetCredential, assetMetadata\}$$

$$assetContext = \{UID, Issuer_{ID}, Issued_{time}\}$$

$$assetCredential = \{Resource_{id}, Domain_{id}, Resource_{type}, Resource_{func}\}$$

$$assetMetadata = \{Resource_{URI}, Resource_{location}\}$$

Asset context information represents the system-related information such as the unique identifier (UID), the issuer ID, and the creation time. Asset credential contains the constant resources information, including the resource ID (granted from the IDM component), domain ID, resource type (e.g., sensor, actuator, tag), and resource function
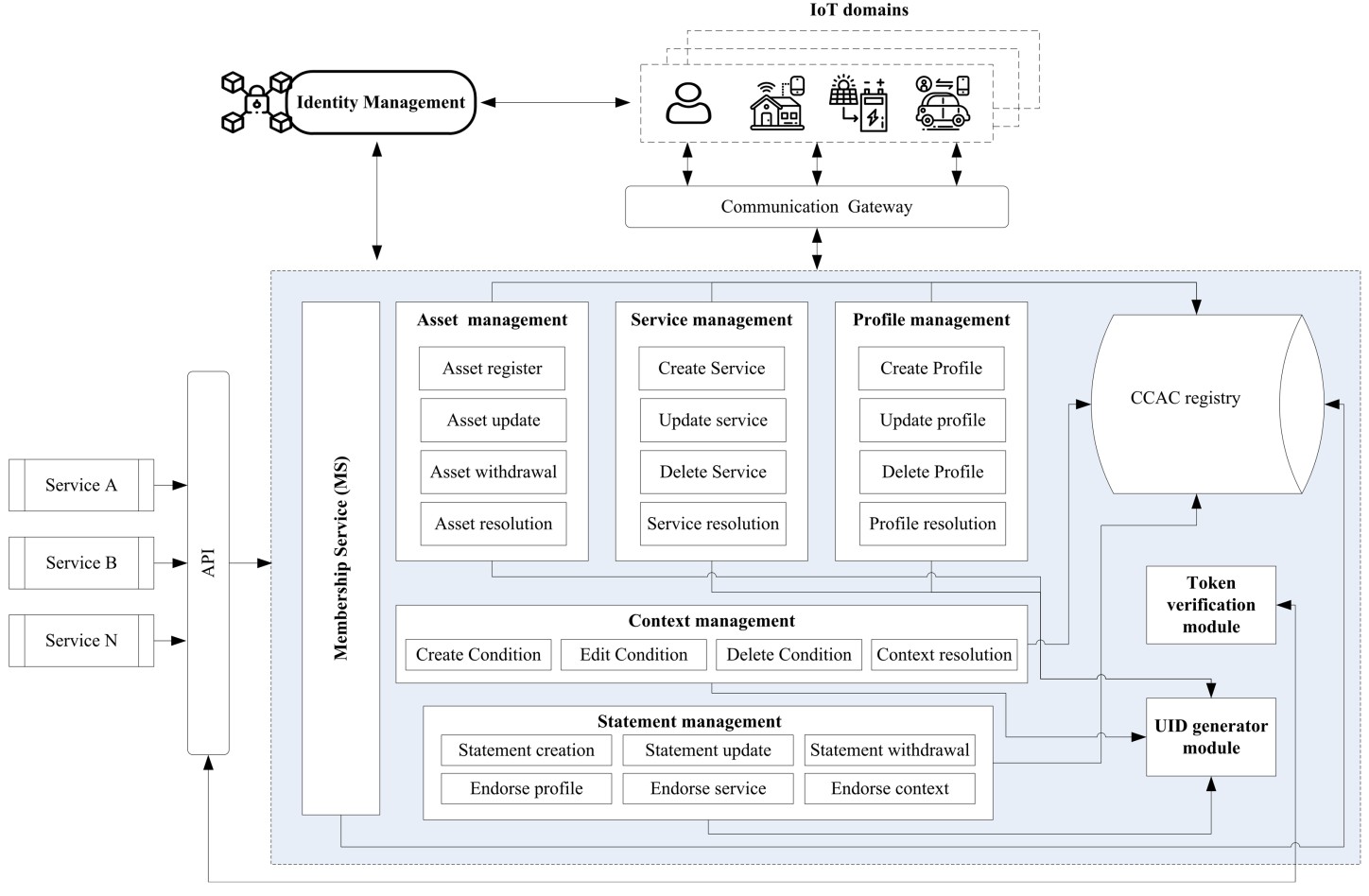

**Figure 1** IoT-CCAC system architecture.

(e.g., temperature, pressure, light). Asset metadata covers the changeable resource information such as resource URI and resource location.

### Service management

Since a consortium may have multiple collaboration projects, each project is interpreted as a service inside the network. The service management module is responsible for creating, editing, and altering service-related operations. Introducing the notion of service to the network will enhance the flexibility and the granularity of the system and regulate the collaboration project's requesters and requests. Properties needed for creating service can be expressed with the following notations:

$$Service = \{serviceConext, serviceCredential, serviceMetadata\}$$

$$serviceContext = \{UID, Issuer_{ID}, Issued_{time}\}$$

$$serviceCredential = \{Service_{name}, Service_{initiator}, Service_{participants} :$$
$$\{domain_1, domain_2, ...domain_n\}\}$$

$$serviceMetadata = \{Requester_{number}, Request_{number}\}$$

Service context represents a service's information in the system, including its unique identifier (UID), issuer ID, and issuance time. Service credential includes but not only a service name, service initiator, and service participants which are a list of participating domains. Service metadata contains regulation and security information, such as the maximum number of requesters and requests.

### Profile management

Conceptually speaking, a profile represents the context information that a physical resource holds in a particular service. One resource may have different profiles, but each profile is defined for only one resource in a particular service. A profile can be assigned to one or multiple statements, and it stands as the resource identifier. The profile management module is responsible for creating, editing, and altering profiles. The alias profiles are represented as follow, where profile context contains the system-related information, and profile credential is defined by corresponding asset ID and the service ID.

$$Profile = \{profileConext, \ profileCredential\}$$

$$profileContext = \{UID, \ Issuer_{ID}, \ Issued_{time}\}$$

$$profileCredential = \{Asset_{UID}, Service_{UID}\}$$

### Context management

Context management is a crucial point of managing access rights as it is the part of defining environment conditions to allow access under some circumstances and denied them under others. Conditions can be location, time, security level, authentication status, protocol, and more. The context information values are gathered from the network resources and the surrounding environment regularly to ensure the correctness of the condition values. The context conditions can be attached to profiles, assets and services metadata to deny or allow access according to the fulfillment of conditions. Context management is presented in the following notations:

$$Condition = \{conditionConext, conditionMetadata\}$$

$$conditionContext = \{UID, \ Issuer_{ID}, \ Issued_{time}\}$$

$$ConditionMetadata = \{Condition_{check(1)}, Condition_{check(2)}, ..., Condition_{check(n)}\}$$

Condition context represents the information of a condition in the system and condition metadata covers the different condition to check before granting access to a

requester. In order to check a condition with the gathered data, we apply the context check function that takes a Boolean format as follow:

$$Condition_{check} = \langle Condition_{constant} \rangle \langle OP \rangle \langle Value \rangle$$

$$Condition_{constant} \in \{Location, Time, Protocol, ...\}$$

$$OP \in \{\geq, \leq, =, \neq, ...\}$$

### Statement management

A statement is a document holding the permission and access rights of a particular resource in a particular service. Statements can be granted to a particular subject or a group of subjects in the form of tokens for access authorization. The statement management module is responsible for registering, updating, and altering statements and also checking the legitimacy of other system information such as profiles and services before each registration or updating operation.

The complete statement definition in IoT-CCAC can be expressed with the following notations:

$$statement = \{statementConext, statementCredential, statementMetadata\}$$

$$statementContext = \{SID, Issuer_{ID}, Issued_{time}, Principal\}$$

$$statementCredential = \{Profile_{ID}, Action, Resource_{URI}\}$$

$$statementMetadata = \{Condition(1)_{ID}, Condition(2)_{ID}, ..., Condition(n)_{ID}\}$$

A brief description of statement elements as follows:

- **SID**: unique identifier for each statement in the system.
- **Issuer**: the issuer of the statement (e.g., service admin).
- **Issued-time**: represent the time of creating or updating the statement.
- **Principal**: for each statement alteration, a new statement will be created and the principal field will have the previous SID value. In the case of first-time creation, the principal field will have the same SID field value. It is mainly used for traceability concerns.
- **Profile**: represents the resource profile in a particular service.
- **Action**: represent the set of access rights that are granted in the statement. Its value could is defined as follow:

$$Action \in \{Read, Write, Read\&Write, NULL\}$$

If Action=NULL, permission denied.

- **Resource_URI**: a URI format used to identify the access path of a particular entity. Represented as follow:

$$Resource_{URL} = Domain_{ID} : Service_{ID} : Region_{ID} : Resource_{ID}$$

Domain ID represents the organization holding the ownership of the entity; service ID represents the application where the entity participates, region ID represents the location of the entity, and the resource ID represents the resource for which the action is granted.

## IoT-CCAC membership service

IoT-CACM Membership Service (MS) implements accounts to interact with its management module. Each account belongs to one domain, and there are two types of accounts consisting of a collection of permission. The first type is administrators that carry full permission to create and alter assets and services related information and assign members to services. The other is service members with the right to perform various network-related operations, such as creating and altering statements, granting access tokens to subjects, and auditing or analyzing reports. Subjects (requesters) simply use client-server abstractions to interact with the access control system after receiving a valid authentication token from the IDM. As a result, the device to device communication is enabled as a resource (asset) in the system can interact with another resource as it holds a valid identity issued by IDM and can request access permission as a standard subject.

## IoT-CCAC token operations

In this subsection, we discuss the capability token operations, starting from converting a statement to a Capability token then the generation of group token then the revocation process.

### *Issuing capability token*

Figure 2 illustrates the system interactions between the subject, IDM, and access control for generating the capability tokens. As an initial step, after defining the elements of the access control and linking the resources and services following the previous steps, all the subjects requesting access must first register to the consortium network via the IDM for a valid identity. Once the subject is successfully registered, it can request a token containing access rights to access a network resource. Further, the service member checks the subject legitimacy and checks if the statement containing the same permissions exists. If it does not exist, SM creates a statement containing the granted access right and the access conditions, as well as filling other statement information as mentioned above. Once the statement is formed, the system creates a capability token using a token generation algorithm and communicates it to the requester following this notation:

$$Cap_{Token} = \{Subject_{ID}, Statement_{ID}, Valid_{Time}\}$$

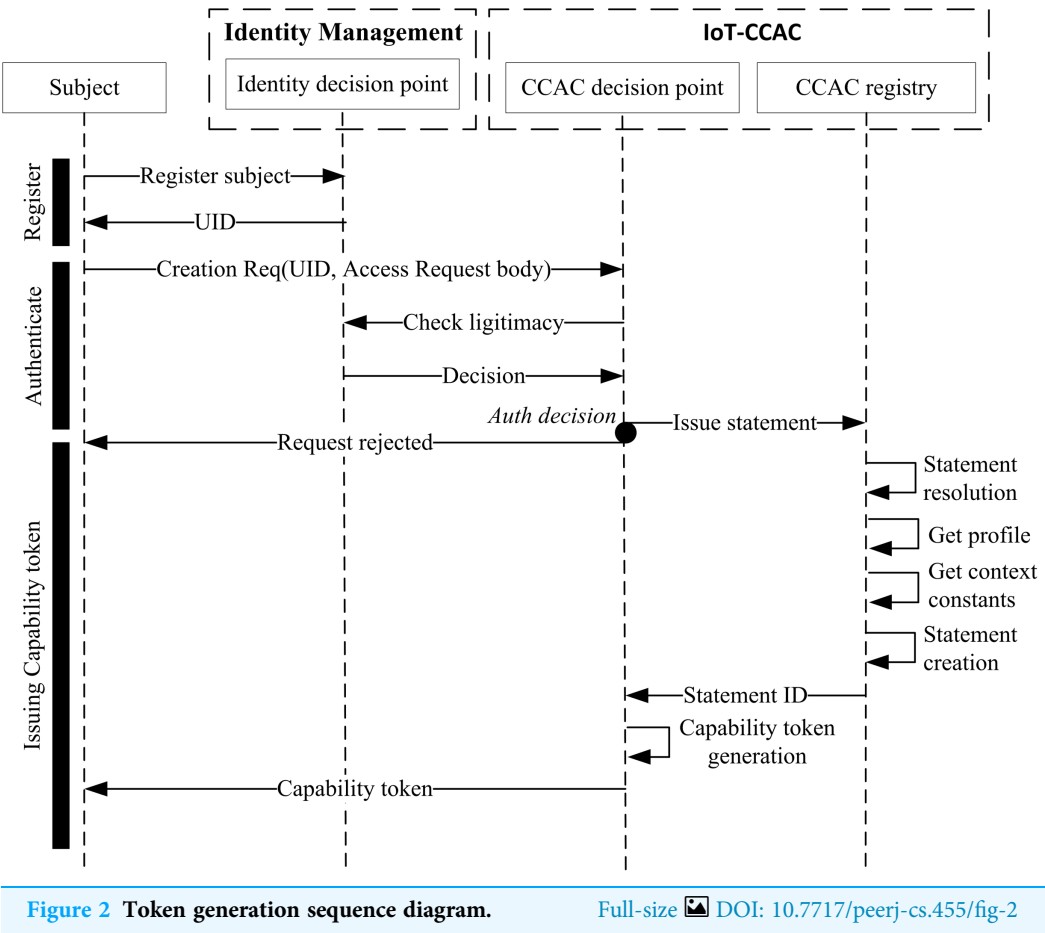

**Figure 2  Token generation sequence diagram.**

$$Valid_{Time} = \{Start_{VT}, End_{VT}\}$$

### Group capability token

Introducing group capability token will help categorize and consolidate the access forms where a group is created and hold few subjects seeking the same access right and access purpose. In our design, a group capability token is supported as we do not store capabilities internally. By design, the statements can be shared among different subjects if they are from the same service and request the same access rights. A subject (group manager) needs to create a valid group identity (GID) from the IDM and send a request containing the GID and other access rights. At the same time, the system will generate a token following the notation:

$$Cap_{Token} = \{Group_{ID}, Statement_{ID}, Valid_{Time}\}$$

### Revocation of capability token

The basic way of revoking a capability token is to store the capability token in a database and perform a simple delete action and check all tokens for every access request. Alternatively, token revocation can be done by adding the token to an exception list and

perform a check task for that list each time a subject sends an access request. In our design, we opted for an exception list to revoke the tokens. For instance, our granular design allows denying access to resources at various levels. Suppose a profile is deleted or a service is archived, or a statement document is altered. In that case, the statements containing outdated data will not be valid when performing an authorization decision task, and the request will be rejected.

### IoT-CCAC authorization process

Figure 3 shows the flowchart of the authorization decision process. The components participating in the authorization decision are IDM and the access control module. IDM is responsible for checking the legitimacy of the subject requesting access.

The IoT-CCAC authorization involves checking the validity of the token, the action granted, the availability of the asset, and the fulfillment of conditions:

- *Check the validity of the token*: the first step of the authorization process is to check the validity of the token. If the token is valid, it will be decoded and the subject ID sent to identity management to check the legitimacy. If the token is not valid or the subject is not authenticated, the request is rejected.
- *Check the approval of access right*: checking if the access method requested matches the access right granted in the statement credentials. If not met, the request is rejected.
- *Check the availability of the asset*: using the profile ID, we check the existence of the profile and services and the availability of the asset. In the case of an unavailable asset, the request will be rejected.
- *Check the fulfillment of conditions*: the last step is to check if the conditions of the statement metadata are fulfilled and match the records on the database. If the condition is met, the request is authorized.

## IOT-CCAC AND BLOCKCHAIN INTEGRATION

This section discusses the different points of choosing decentralized data registry architecture over a centralized architecture for an IoT access control system based on the requirements of the city, business, and utilities IoT application (*Abou Jaoude & Saade, 2019*).

### IoT access control data registry requirements

The data layer of access control is a critical component and the most vulnerable as it persistently stores the necessary data. The system acts upon the stored information to answer the correctness of the operations. For instance, by nature, IoT is decentralized as each domain owns a sub-network of objects, and the IoT network is predicted to be the network of billions of sensors and connected devices, which will require high reliability, and availability, to support such network. Besides, the crucial element to unlock the value of IoT is the interoperability and data exchange between the sub-networks; henceforth, integrity, confidentiality, and transparency are crucial to achieving the purpose (*Yaqoob et al., 2017*).

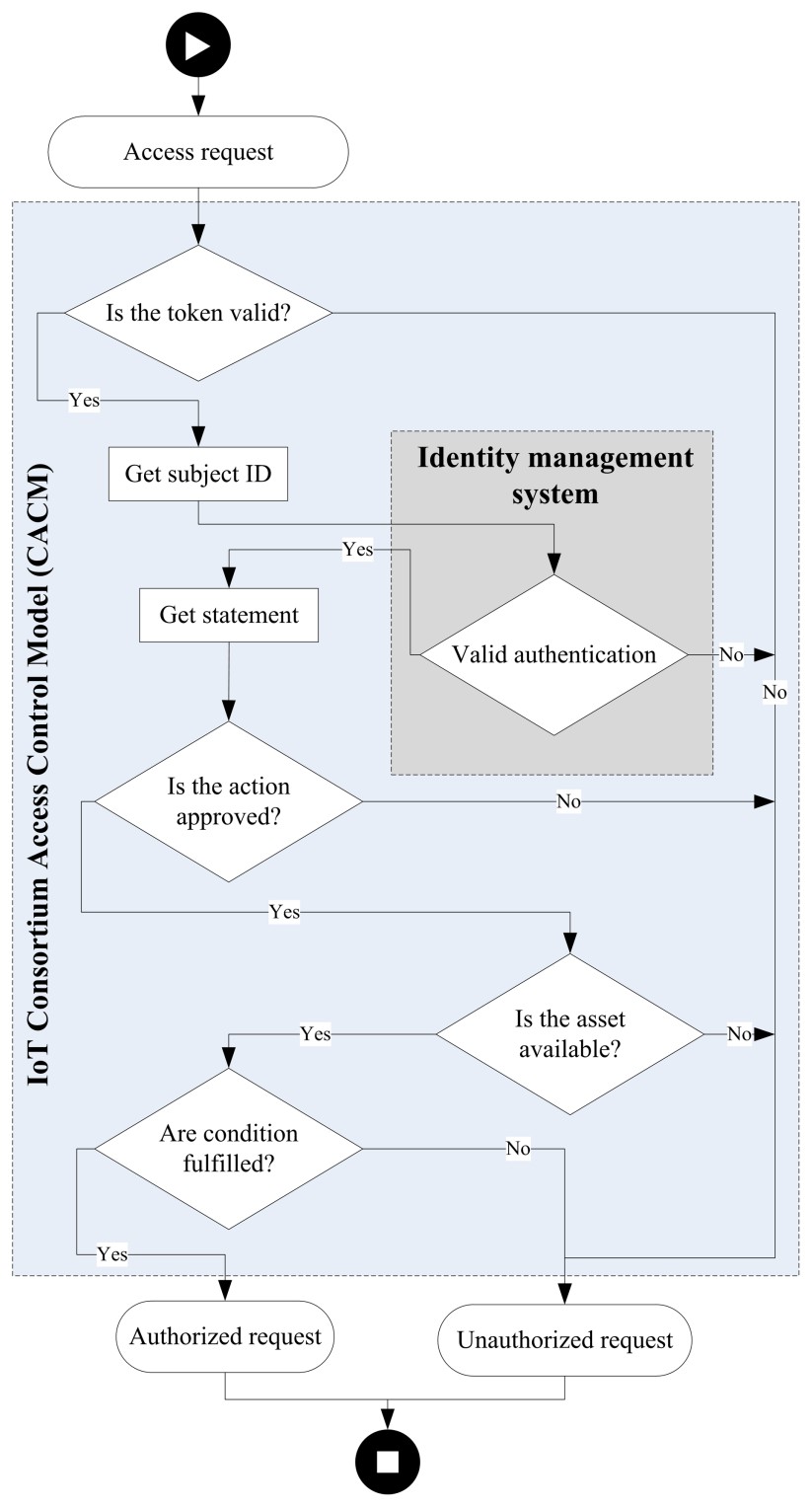

**Figure 3** **Authorization process.**

**Table 3 Comparison between traditional database and blockchain for IoT access control (*Tseng et al., 2020*).**

| | | Database | | Blockchain | |
|---|---|---|---|---|---|
| | | Advantages | Disadvantages | Advantages | Disadvantages |
| IoT access control security requirements | Reliability | – | Data mutability | Data immutability | Data mutability |
| | Availability | Data mutability | Single point of failure | Decentralized architecture/ fault tolerance | – |
| | Integrity | – | Centralized authority/ data mutability | Data immutability/ transaction conformation | – |
| | Confidentiality | – | Centralized authority/ exposed data | Owner-controlled data/ encrypted data | – |
| | Transparency | – | Centralized authority | Decentralized authority | – |
| IoT access control performance requirements | Performance | High transaction performance/low latency | – | – | Low transaction performance/high latency |
| | Scalability | High Scalability | – | – | Scalability comes with price |
| | Capacity | Easy to run at any capacity | – | – | Resource and energy- intensive consumption |
| | Usability | Easy to use and to deploy | – | – | Configuration of different components |
| | Maintenance | Low cost | – | – | High cost |

A blockchain is an immutable digital ledger formed by blocks that uses cryptography practices to store data. It can provide properties such as decentralization, immutability, and enhanced security, while traditional databases allow data to be stored in different data structures such as tables or documents with properties of competent transaction performance, scalability, usability, and low-cost maintenance. Table 3 shows the advantages and disadvantages of adopting blockchain and a traditional database to meet the security and performance requirements of the IoT access control system.

In the final analysis, blockchain meets the access control security requirements, and the database leverages performance. The Blockchain technology was created to support the concept of decentralized monetary systems such as Bitcoin and Ethereum, where the databases are better for system performance as they have been used since the early age of creating computers. Our purpose is to deliver a secure, robust access control system to meet IoT domains' needs and leverage IoT value by enabling interoperability and data exchange. Using the traditional database to backbone the IoT-CCAC will certainly leverage a robust solution; many research works and existing enterprise incidents have already proven that security issues will arise, such as data breaches, single-point access, and the lack of transparency. For this reason, we adopt blockchain-based database technology to enhance the security of IoT-CCAC.

## Blockchain integration

Figure 4 shows the IoT-CCAC based blockchain architecture, which consists of a consortium network (IoT domains), IoT consortium capability access control module, and blockchain-based database registry. The consortium is formed by the members participating in the network to achieve a business goal or collaborate in a particular project.

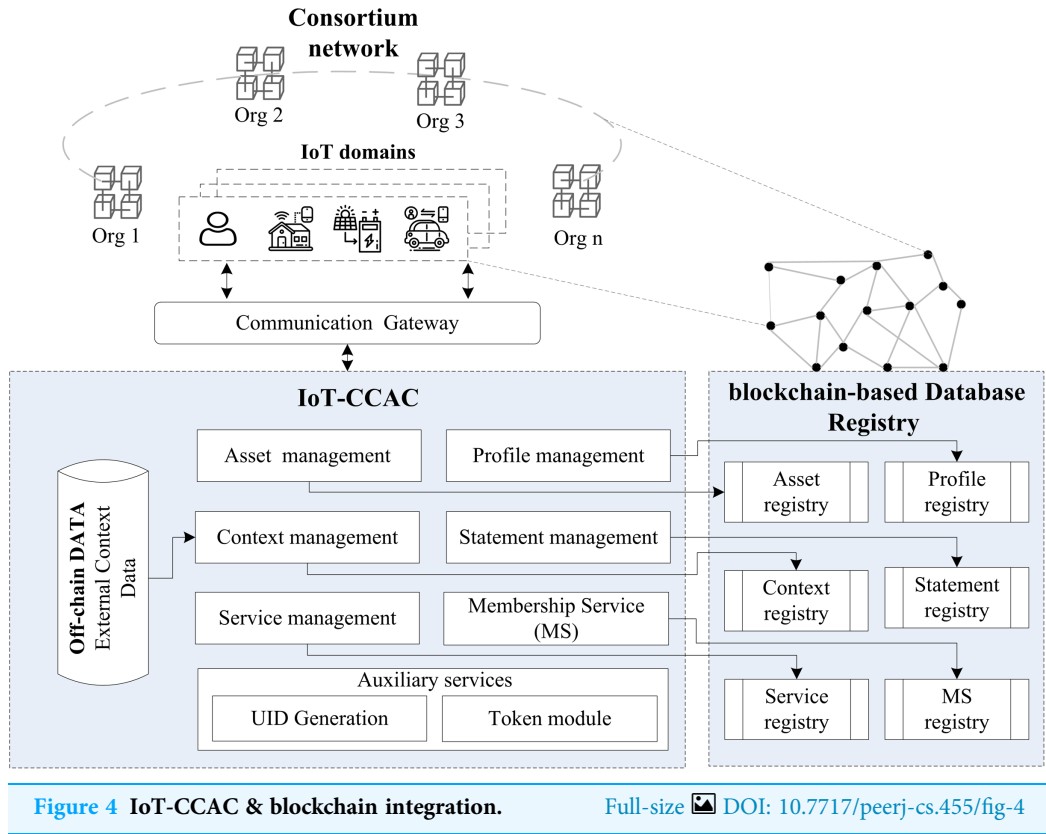

**Figure 4  IoT-CCAC & blockchain integration.**     

Every member needs to provide a node or more to participate in the network operations and hold a copy of the data.

The IoT-CCAC module were explained in the previous section. Each module connects and interact with its registry. The off-chain data store is a standard database that stores the environment data coming from the devices and sensors in the network and participates in context management checking. And the blockchain-based database is a hybrid solution that assembles the security characteristics of blockchain and the database performance in one data registry. Adopting a blockchain-based database for IoT-CCAC will bring all the database properties such as high transaction rate, data indexing, and querying, and friendly usability, and will enhance the security of the access control data registry by making it resistant to unauthorized changes without a need for any trusted third party to answer the integrity or the confidentiality of the registry, as all the consortium members hold a copy of the data and it is maintained by cryptography practices.

## Use case scenario

To better explain the proposed architecture as well as its integration with blockchain technology and evaluate its feasibility, we turn to the use case scenario of waste management in a smart city.

In the context of our IoT-CCAC, say that there are three organizations working on a collaborative project (a service) of waste management. In this service, the city council

oversees the whole process and manages the garbage can sensors; the recycling plant is responsible for sorting the recycled garbage; and the manufacturing plant then processes the classified recycled materials to manufacture specific products. The sensors and devices recording relative data belong to different organizations. They are considered their respective assets inside the system, and each asset can possess more than one profile considering it can participate in other services, and the processes of registering services and creating digital assets for a particular organization is by holing an account in MS.

Given the waste management service, when a supervisor from the city council needs to read all project-related data to have a clear picture of the current status of the whole project, registration and authentication through IDM is needed. He will then need to request a capability access token. A capability token (a statement inside a system) is granted to a supervisor only if he satisfies the system requirements. Using the acquired token, the supervisor can authenticates and send an access request to corresponding asset. On receiving an access request containing the capability token, the token authorization process will decide if the access is granted or not after performing all the checks.

## IMPLEMENTATION AND EVALUATION

In this section we will discuss the implementation stages and the evaluation results. Firstly, we discuss the system design, present the testing environment, the employed technologies, and finally we discuss the obtained results.

### System design discussion

- *Who is going to use it?*
  Our system is designed to fit the categories of business, utilities, and enterprise domain applications where several organizations (domains) want to cooperate and share their resources for defined projects (services) to enable the potential of IoT sharing applications such as smart city paradigm.
- *What are the requirements of the system?*
  In this IoT application scale, reliability and availability are essential as the services' intelligent decisions are based on the vast amount of data continually collected from the network resources. Confidentiality and integrity are secondly important as any compromised data might lead to a wrong decision that will impact the consortium business plans and objectives. End-user privacy is not much required in such applications as they are not potentially involved in the interaction with the system (end-user privacy involves in the personal network).
- *Who are the users of the system?*
  Our system is controlled by a set of administrators where each domain has an account, and each account has the role of an administrator. In the same manner, an account also has a service member role responsible for the management of services inside the system. Interacting with the system is done by a simple client-server abstraction without using any system-related notion or task.

- *What are the inputs and outputs of the system?*
  The input data of our system are the physical resources registered in the form of assets and the data gathered from the environment to be consumed in the condition fulfillment process. On the other hand, the output is a payload object that contains an authorization decision.

## Experiment environment

We evaluated our solution by simulating the use case scenario of waste management in the previous section. For instance, we simulated three organizations collaborating on several services, each organization can register physical devices as assets and generate Json Web Tokens matching the access control statements. Our experiment results is based on two types of the data store; first one is implemented locally (offline) using Docker technology and the second one we use the BigchainDB online test node (https://test.ipdb.io/). The different components of the experiment use RESTful API to exchange data.

## Experiment setting

In order to examine the performance of our proposed access control solution we implemented our prototype using Python programing language, FLASK micro web framework, and JWT Crypto Library. We employed BigchainDB, a blockchain-based database as the data store node using Docker container. BigchainDB node contains BigchainDB 2.0 server, a mangoDB database and a Tendermint as consensus protocol. The execution environment is a virtual machine running xUbuntu with 4 GB of RAM and 1 CPU Intel Core i7-4510U 2.00 GHz. We also used Apache JMeter to simulate simultaneous registration and authentication requests.

## Security analysis

To evaluate the security of our solution we present several common attacks in the decentralized system and we discuss our approach to avoid such attacks.

- Forgery attack: it's a common attack of tampering identities and transaction data to get access to confidential information or pollute the system with random data.
- Injection Attacks: an attacker can inject a script to manipulate the authorization process or to alter a database record or to carry out an unwanted action.
- Man in the middle attack: it's when the attacker secretly stands in the middle between two communicating entities and read the exchanged data.

  We prevent such attacks by implementing the following preconditions:

- The assets identities are unknown to attackers and to other participating organization as we only exchange externally the capability tokens corresponding to statements which contains the profile ID not the asset ID.
- We use SHA256 algorithm to digitally sign the exchanged messages and tokens which makes it hard to forge or to alter.

**Table 4 Computing and communication cost for each system transaction (average time of 100 transactions is presented).**

| Transaction type | Preparation Time (ms) | Fulfillment Time (ms) | Commit Time Offline (ms) | Commit Time Online (ms) |
|---|---|---|---|---|
| Asset | 1.2 | 2 | 110 | 1,210 |
| Service | 1 | 2.4 | 110 | 1,170 |
| Profile | 1.5 | 3 | 110 | 640 |
| Statement | 1 | 2.6 | 110 | 900 |

- For each system input we run different checks to ensure the legitimacy of the information before accessing the data store.
- Adopting blockchain technology is another strong point to enhance the security of the system and to prevent forgery attacks.

## Experiment results and discussion

To verify the effectiveness of IoT-CCAC, we conducted several test experiment, firstly we calculate the communication and computation cost for creating assets, profiles, services, and statements using the local data store and the online testing node. The results are presented in the Table 4. Transaction in BigchainDB flows in two stages before committing it for permanent storage.

- Preparation stage: the stage of constructing the transaction and executing initial input checks to ensure the validity of the transaction. At this stage the size of the testing transaction is 240 bytes.
- Fulfillment stage: the stage of signing the transaction with the creator private key and hash its body content to be the ID of the transaction. At this stage the size of the testing transaction is 368 bytes.

The second experiment is to send bulk transactions to the server to test the performance and the scalability of the data store in term of handling concurrent transactions. Using apache JMeter, We piloted 4 groups of 10, 50, 100, 200 concurrent transactions for create and authenticate operations. The Fig. 5 shows the execution time of creation operation, and the Fig. 6 shows the execution time of authentication operation. The $x$-axis present the execution time (in millisecond), and the $y$-axis presents the 4 bulk transactions group and the series represents the average time of the transaction commit, transaction latency, and time to connect the server. From the first sight we can see that creation operation takes more time as a transaction have to accomplish two verification steps before writing it inside a block. For instance, when the BigchainDB server receives the creation transaction it will check the legitimacy of the transaction by verifying the signature of the issuer and the correctness of the data by hashing the transaction content and comparing it to the transaction ID; if both checks are valid and the transaction is not a duplicate inside the system the transaction will be written inside the blockchain database. The authentication operation is relatively faster as we take the advantage of database

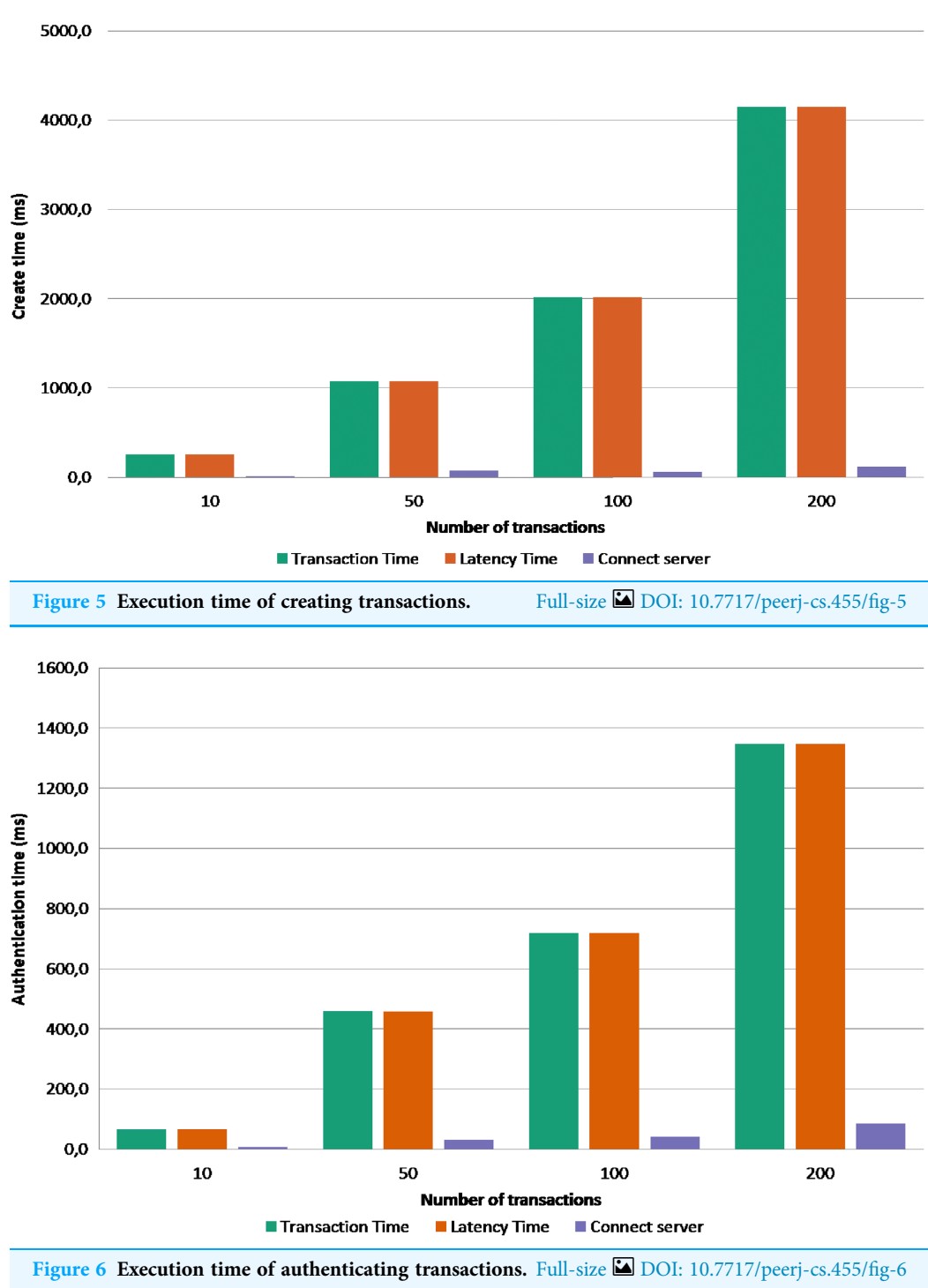

**Figure 5 Execution time of creating transactions.**

**Figure 6 Execution time of authenticating transactions.**

fast querying; we only check the requester signature and fetch the different transactions using their ID's. Figure 7 shows the accumulative latency time of 50 simultaneous authentication request. The more requests reach the server the latency time is longer. As a solution, a vertical or horizontal resource scale will reduce the latency time and reach the wanted performance.
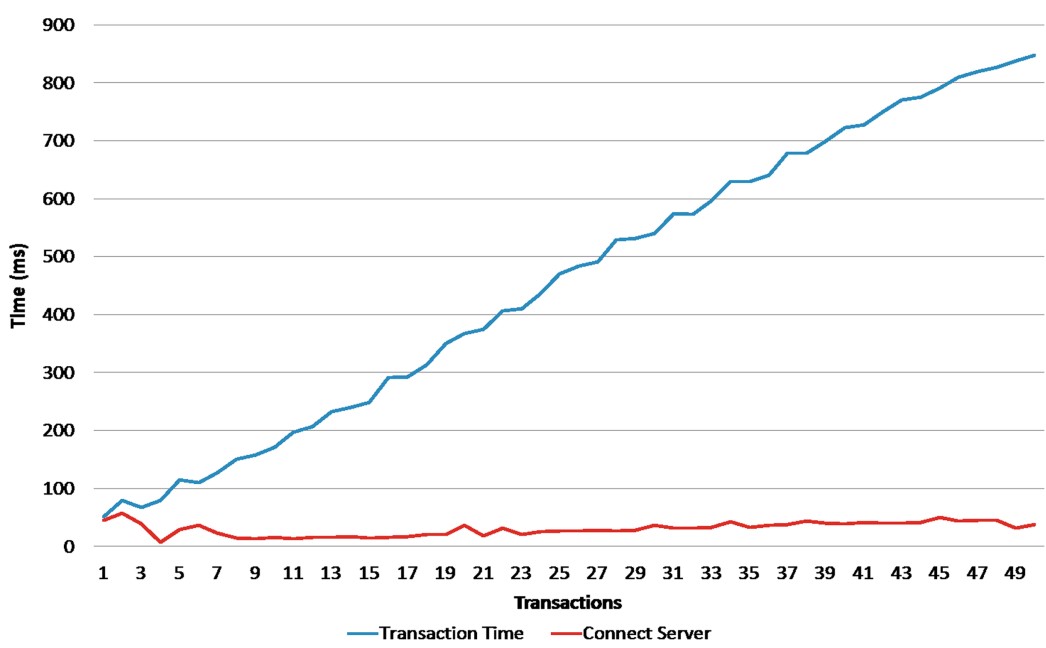

**Figure 7** **Latency time of 50 simultaneous authentication transactions.**

   Our solution showed a better performance results compared to related work as we adopted the blockchain-based database technology to benefit of the blockchain security properties and the high performance of the database. From the experiment results it can be observed that our solution can achieve the performance needed for IoT city-level access control. In addition, the flexibility, and the interoperability of IoT-CCAC makes it adoptable for different use cases and IoT applications.

## CONCLUSION

In this paper, we presented a blockchain-based consortium access control approach for IoT large-scale applications. We first compared the capability access control model (CBAC) to the role and attributed based access control (RBAC, ABAC) and highlight the advantages of adopting CBAC over the others for IoT applications. In the architecture design, we presented a novel concept of managing the access control data to enable flexibility, interoperability, and data exchange between the consortium members.
We explained the system assets, services, profiles, statements, membership service, and the token generation protocol, including the authorization process. Secondly, we discussed the IoT access control data store requirements, and we conducted a comparison between blockchain security features and database performance properties. We explained the benefits of adopting a blockchain-based database as the IoT-CCAC data store and discussed its integration architecture. A concept-proof prototype was implemented and evaluated in terms of security and performance to verify the feasibility of IoT-CCAC. Our IoT-CCAC approach showed promising results and a good fit for city and business network applications.

Despite our approach's encouraging results, a part of our ongoing efforts is to investigate and further explore the blockchain-based database security and privacy for access control in IoT networks and application scenarios.

### Funding

The authors received no funding for this work.

### Competing Interests

The authors declare that they have no competing interests.

### Author Contributions

- Mohammed Amine Bouras conceived and designed the experiments, performed the experiments, performed the computation work, prepared figures and/or tables, and approved the final draft.
- Boming Xia conceived and designed the experiments, performed the experiments, prepared figures and/or tables, and approved the final draft.
- Adnan Omer Abuassba analyzed the data, authored or reviewed drafts of the paper, and approved the final draft.
- Huansheng Ning analyzed the data, authored or reviewed drafts of the paper, and approved the final draft.
- Qinghua Lu analyzed the data, authored or reviewed drafts of the paper, and approved the final draft.

### Data Availability

Data is available at GitbHub: https://github.com/mohamine18/CCapAC.

### Supplemental Information

Supplemental information for this article can be found online at http://dx.doi.org/10.7717/peerj-cs.455#supplemental-information.

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
