# Peer review of "IoT-CCAC: a blockchain-based consortium capability access control approach for IoT"

_PeerJ Computer Science, doi:10.7717/peerj-cs.455_

## Round 0.1 · original submission · Major Revisions

· Academic Editor

Major Revisions

The current findings are based on some preliminary results. More experiments should be done to better support the conclusion. Codes should be provided for verification of results.

Reviewer 1 ·

Basic reporting

In this paper authors proposed IoT-CCAC based blockchain framework and results showed that IoT-CCAC approach gives promising results and a good fit for city and business network applications.

The paper is well written and easy to follow, however the novelty of the paper is limited.

Experimental design

Authors presented detailed description of IoT-CCAC module but blockchain details are missing. Not sure which platform of blockchain is used by authors and why they chose only that. They have mentioned two blockchains (Ethereum private blockchain, Hyperledger Fabric).

Validity of the findings

The python code of the experiment is not attached and therefore hard to verify results. I would suggest either to publicly upload the code and instructions on github or attach here as supplementary file.

Additional comments

Overall the paper looks fine and well written. But it is important to verify the results presented by authors and it is only worth to accept the paper if experiments can be verified. Therefore I would suggest a major revision with above comments.

Reviewer 2 ·

Basic reporting

The paper discusses the utilization of the blockchain technology for IoT access control.
The general presentation of the paper is good which makes it very to read since the main contributions are clearly highlighted.

I suggest adding some references to Table 1 and 3.

Experimental design

no comment

Validity of the findings

no comment

Additional comments

The authors presented in Table 3 a list of the classical DB's disadvantages. For instance the mutability, the point of failure,. ... etc. However what would be the general thoughts in the case of the use of a nosql DB (or any other big-data implementation) where the database management system ensures a high degree of redundancy. and recovery.

·

Basic reporting

While the article includes tables and graphs representing experiment results, authors do not provide raw data.
Experiment methodology should be described in greater detail, as further explained in the second part of this review.
Authors use clear and technically correct English, provide relevant references and article is structured in an acceptable format.

Experimental design

It is not clear if the measurement results presented were obtained from only one measurement or if they are an average of multiple runs. The label of Table 4 states numbers presented are an average of 100 transactions and lines 372-274 that four groups of 10, 50, 100 and 200 concurrent transactions were used. However, it is unclear if measurements for these four groups were repeated or run only once. The initial state of both local VM and online node used for experiment can have an impact on the accuracy of the results.
Security analysis is superficial, and although preconditions presented improve IOT-CACC model security, attack prevention mechanisms are not exhaustive. The statement on lines 359-360 does not specify what checks are made or even references the IOT-CACC authorization process provided in system architecture.

Validity of the findings

Findings presented are based on preliminary results and make it difficult to call them robust. Due to deficiencies mentioned in the second part of this review, it is hard to compare results with related work conclusively. As such, the statement on lines 389-391 requires more experiments for direct comparison.

Additional comments

IOT-CACC model presented is promising, but further investigation to support presented conclusions is required.

---

## Round 0.2 · accepted · Accept

· Academic Editor

Accept

I am writing to inform you that your manuscript - IoT-CCAC: a blockchain-based consortium capability access control approach for IoT - has been Accepted for publication.